# Laser Flare Photometry to Monitor Childhood Chronic Uveitis: A Preliminary Report of a Monocentric Italian Experience

**DOI:** 10.3390/diagnostics13203179

**Published:** 2023-10-11

**Authors:** Ilaria Maccora, Cinzia De Libero, Matilde Peri, Gioia Danti, Alessio Rossi, Edoardo Marrani, Roberta Pasqualetti, Ilaria Pagnini, Maria Vincenza Mastrolia, Gabriele Simonini

**Affiliations:** 1Rheumatology Unit, ERN ReConnet Center, Meyer Children’s Hospital IRCCS, 50139 Florence, Italy; edoardo.marrani@meyer.it (E.M.); ilaria.pagnini@meyer.it (I.P.); maria.mastrolia@unifi.it (M.V.M.); gabriele.simonini@unifi.it (G.S.); 2NeuroFARBA Department, University of Florence, 50121 Florence, Italy; 3Ophthalmology Unit, Meyer Children’s Hospital IRCCS, 50139 Florence, Italy; cinzia.delibero@meyer.it (C.D.L.); gioia.danti@meyer.it (G.D.); roberta.pasqualetti@meyer.it (R.P.); 4School of Health Human Science, University of Florence, 50121 Florence, Italy; matilde.peri@unifi.it (M.P.); alessio.rossi@unifi.it (A.R.)

**Keywords:** uveitis, children, eye, laser flare photometry, juvenile idiopathic arthritis

## Abstract

**Background**: Childhood chronic non-infectious uveitis (cNIU) is a challenging disease that needs close monitoring. Slit lamp evaluation (SLE) is the cornerstone of ophthalmological evaluation for uveitis, but it is affected by interobserver variability and may be problematic in children. Laser flare photometry (LFP), a novel and objective technique, might be used in children with uveitis. **Aim**: The aim of this study was to attempt the use of LFP in cNIU clinical practice. **Methods**: Children, attending the Rheumatology Unit and who were scheduled to receive ophthalmological evaluation, were prospectively enrolled to concomitantly receive SLE and LFP. SLE was performed blind to LFP measure. Demographic, laboratory, clinical, and ophthalmology data were collected. **Results**: A total of 29 children (58 eyes) were enrolled, including 3 with juvenile idiopathic arthritis without uveitis (JIA-no-U), 15 with JIA-associated uveitis (JIA-U), and 11 with idiopathic chronic uveitis (ICU). We observed significantly higher LFP values in the eyes of children with uveitis compared to the others (10.1 IQR 7.1–13.6 versus 6.2 IQR 5.8–6.9, *p =* 0.007). Accordance between the SLE and LFP measures, at baseline (ρ.498, *p* < 0.001) and during the follow-up (LFP II ρ 0.460, *p* < 0.001, LFP III ρ 0.631, *p* < 0.001, LFP IV ρ 0.547, *p* = 0.006, LFP V ρ 0.767, *p =* 0.001), was detected. We evaluated significant correlation between LFP values and the presence of complications (ρ 0.538, *p* < 0.001), especially with cataract formation (ρ 0.542, *p* < 0.001). **Conclusions**: In this cohort, LFP measurements showed a good correlation with SLE. LFP values showed a positive correlation with the presence of complications. LFP might be considered as a reliable objective modality to monitor intraocular inflammation in cNIU.

## 1. Background

Chronic non-infectious uveitis is a challenging disease from a diagnostic and treatment perspective, due to the risk of severe complications, including blindness, if it is not promptly and correctly identified and treated [1,2,3,4]. Uveitis, according to the Standardized Uveitis Nomenclature (SUN), is defined based on the anatomical location, the disease course, and the clinical symptoms [5]. In childhood, idiopathic uveitis and juvenile idiopathic arthritis (JIA)-associated uveitis account for the majority of cases of chronic uveitis, with typical involvement of the anterior segment [1,2,6].

Childhood chronic non-infectious uveitis (cNIU) is frequently asymptomatic and eventually develops into severe ocular complications and visual loss. Persistent ocular inflammation and chronic use of corticosteroids can lead to the development of severe ocular complications including cataracts, glaucoma, posterior synechiae, and band keratopathy [1]. In order to prevent these complications and therefore blindness, it is crucial to start proper systemic treatment and perform close monitoring. In this vein, the slit lamp examination (SLE) allows evaluation of the number of cells present in the anterior chamber and aqueous flare which are the direct consequence of exudation for inflammation. The strength of cells, and protein exudation, directly correlate with the severity of inflammation and are graded based on the SUN classification [5]. In JIA, due to the well-known risk of uveitis, an established screening program with SLE leads to early interception of uveitis development and its complications [7]. SLE represents, therefore, the cornerstone of ophthalmology evaluation for uveitis in JIA, and overall cNIU [1,6].

However, this is a subjective technique that is burdened by interobserver variability. Additionally, it is not able to intercept subclinical changes in the blood-–aqueous barrier that can start in the earliest phase, and it may also easily miss a low number of cells, mirroring a persistent, sub-acute/chronic inflammation, thus eventually leading to complications. Additionally, in children, SLE might be challenging because of scarce collaboration.

In the last few years, increasing progress has been achieved in quantifying intraocular inflammation in an objective way, and one of them is laser flare photometry (LFP). LFP is an objective, reliable, quantitative, and non-invasive technique that measures the light scattering of a helium–neon laser beam in the anterior chamber based on the concentration of proteins [8,9,10]. These measurements are directly determined by the activity of uveitis because of the integrity of the ocular blood–aqueous barrier [9].

Our aim was to compare the LFP measurements with AC cell grading by SLE according to the SUN classification.

## 2. Materials and Methods

### 2.1. Study Design and Setting

This was a prospective observational study of JIA and cNIU children currently followed at the rheumatology and ophthalmology units of Meyer Children’s Hospital IRCCS. They underwent LFP assessment during a scheduled ophthalmology evaluation by SLE, between 15 February 2022 and 31 May 2023.

### 2.2. Population in Study

All children who fulfilled the following inclusion criteria were included: (A) to be ≤16 years old when LFP was performed, (B) to undergo LFP measurement the same day of the SLE, (C) and to be diagnosed for JIA according to the ILAR definition [11] or cNIU according to the SUN criteria [5]. Patients were excluded from the study if (A) the medical records were incomplete for clinical information, (B) they refused to participate, or (C) if the LFP measurement available was performed within 3 months from surgery, or (D) they refused to undergo LFP. 

Topical corticosteroid and cyclopentolate were permitted.

Ethical approval was obtained from the Institutional Review Board of Meyer Children’s Hospital (RChildUv 27/2022, date of approval 15 February 2022). The research was conducted in accordance with the tenets of the Declaration of Helsinki.

### 2.3. Study Procedures and Data Collection

Children enrolled in the study underwent a standard ophthalmology visit that included evaluation of visual acuity, intraocular pressure measurement, SLE, fundoscopy, complication evaluation, and LFP measurement. SLE was performed, independently, by an expert ophthalmologist (RP), while LFP was performed, blinded, in all the patients, by the same ophthalmologist (CdL) along with always the same technician (GD), as a part of the routine ophthalmological examination at each visit since the enrolment.

Flare values were measured using a Kowa FM-700 laser flare photometer (Kowa Acculas, San Jose, CA, USA), which utilizes a diode laser to measure aqueous flare through a 0.3 × 0.5 mm sampling window. The obtained values are expressed as photon counts per millisecond (ph/ms). Each patient underwent six measurements, with the highest and lowest readings being excluded from the calculation. The average (denoted as AVG) and standard deviation (SD) were automatically calculated based on the remaining four flare measurements.

The following data were collected: -Demographic data including age, sex (female or male), ethnicity, presence of comorbidity, family history of autoimmune disease in a first-degree relative, underlying disease (distinguished as JIA without uveitis, JIA with uveitis and idiopathic uveitis), age at onset (expressed in months), duration of disease (expressed in months).-Laboratory data including ANA, ANCA, HLA-B51, HLA-B27, and inflammatory parameters at onset if available (C-reactive protein (CRP) expressed in mg/dL (normal value < 0.5 mg/dL), and erythrocyte sedimentation rate (ESR) expressed in mm/h (normal value < 10 mm/h)).-Characteristics of uveitis including laterality of uveitis (defined as unilateral or bilateral), anatomical location according to SUN [5] (anterior, intermediate, posterior, and panuveitis), symptoms at onset, as per medical chart, reporting if present or not including the type of symptoms reported (pain, redness floaters, or blurred vision).-Ocular complications were recorded at onset and at each visit when LFP and SLE were performed, and they were recorded as present or absent for each eye. The following complications were recorded: cataracts, elevated intraocular pressure (>21 mmHg), ocular hypotony (<5 mmHg), optic disc swelling, macular oedema defined as intraretinal cysts on optical coherence tomography scan, posterior synechiae, epiretinal membrane, band keratopathy, and choroidal neovascular membrane.-Visual acuity (VA) was recorded at onset and at each visit in LogMAR, and when this scale was not available, the appropriate conversion was performed [12], and VA was stratified as ≤0.3 LogMAR, visual impairment if ≥0.4 and <1, and blindness if ≥1.-Inflammatory status of the eye was evaluated using SLE for the anterior chamber according to the SUN working group grading system defining anterior chamber cells and flare between 0.5+ and 4+ [5]. Fundoscopy and fluorescein angiography were used to evaluate the intermediate segment and the posterior segment of the eye. LFP was defined as normal if the value was ≤5 ph/ms according to previously reported cut-off values [8,9,13,14,15].-The treatments performed at each visit were recorded, including topical as well as systemic treatment.

### 2.4. Statistical Analysis

Statistical analyses were performed using SPSS v28.0 for Microsoft. Continuous variables are described using mean and standard deviation or median and range depending on their distribution. Q–Q plots were used to test the assumption of normality. For dichotomous and categorical variables, proportions were used to assess the clinical characteristics of the population. A *p*-value of <0.05 was considered to indicate statistical significance. Differences among categorical variables were assessed using the chi-square test or the Fisher exact test as appropriate. Specifically, the Fisher exact test was used for categorical variables with 2 categories, when 1 or more of the cell counts was less than 5. Continuous variables were compared with Student’s *t*-test or non-parametric tests such as the Kruskal–Wallis test as appropriate. Spearman’s rank correlations were performed to assess any relationship between baseline and follow-up LFP values and AC cell grades, and the correlation of these parameters with the ocular complications.

## 3. Results

### 3.1. General Characteristics of the Population

We collected the data of 29 children (Figure 1): 3 with JIA without uveitis (10.3%), 15 with JIA-uveitis (51.7%), and 11 with idiopathic chronic uveitis (37.9%) with a median follow-up of 5 months (range 0–16 months). The median age at disease onset was 60 months (IQR 32–138), with a median age at the first LFP measurement of 147 months (IQR 105.25–173.25). Of these 29 children, 19 were female (65.5%), 28 Caucasian (96.6%), 23 had ANA positivity (79.3%), 3 ANCA positivity (10.3%), 1 HLA-B27 (3.4%), and 1 HLA-B51 (3.4%). Additional characteristics of the study population are listed in Table 1.

Among the 26 patients with uveitis, 22 had a bilateral involvement (84.6%), 21 an anterior uveitis (80.7%), and 5 panuveitis (19.2%). Ocular symptoms were reported in 11 patients (42.3%), as pain in 4 (15.4%), redness in 11 (42.3%), photophobia in 6 (23.1%), and floaters/blurred visions in 4 (15.4%). The mean visual acuity at onset was 0.06 (SD ± 0.20) in the whole cohort, while in the JIA-U we observed a mean of 0.07 (SD ± 0.27), and in idiopathic of 0.056 (SD ± 0.13).

At onset, nine patients (31%) showed ocular complications, with a median number of complications of 0 (IQR 0–2.5). The most frequent reported complications at onset were optic disc swelling in 12 eyes, posterior synechiae in 11 eyes, cataracts in 4 eyes, ocular hypertension in 2 eyes, band keratopathy in 1 eye, chorioretinal scar in 1 eye, and cystoid macular oedema in 1 eye.

Among the different groups, we observed that patients with JIA-associated uveitis were significantly younger (*p* = 0.025) and more frequently have anterior uveitis (χ^2^ 8.44, *p* = 0.004), while patients with idiopathic uveitis were less frequently ANA-positive (χ^2^ 6.69, *p =* 0.035), they more frequently had ocular symptoms at onset (χ^2^ 13.87, *p* < 0.001), and more complications compared to the other groups (*p* < 0.001) (Table 1).

### 3.2. LFP Findings in the Whole Cohort

All the 29 patients underwent to at least one LFP measurement at a median age of 147 months (IQR 105–173), after a median duration of the disease of 51 months (IQR 16.5–101.25). The mean value of the first LFP was 14.73 ph/ms (±20.3) in the whole cohort, with significant differences among eyes with uveitis versus eyes without uveitis, according to SLE evaluation (10.1 IQR 7.1–13.6 versus 6.2 IQR 5.8–6.9, *p* = 0.007) (Figure 2) (Table 2).

### 3.3. LFP Findings in Patients with Uveitis

Analysis of LFP values in children with uveitis were then stratified according to uveitis activity and type of uveitis, based on SLE evaluations. At the first LFP assessment, the LFP median values of the eyes of subjects with active uveitis (15 eyes) were significantly higher when compared to the LFP median value of subjects with inactive uveitis (35 eyes), being 12.8 IQR 11.4–44.8 versus 8.7 IQR 5.8–12, *p* = 0.002 (Figure 3). We did not observe a significant difference in the LFP measures regarding the type of uveitis: JIA-U and idiopathic uveitis LFP values were not different (Kruskal–Wallis test, *p* > 0.05).

At the time of the first LFP assessment, the mean AC of subjects with uveitis was 0.29 (±0.66), with 17 eyes with complications (29.3%), of whom 9 had cataracts (15.5%) and 14 posterior synechiae (Table 2). Patients with idiopathic uveitis at the moment of the LFP measurement had a higher prevalence of ocular complication compared to the others (χ^2^ 3.84, *p* = 0.05).

### 3.4. Correlations of LFP Findings with Ocular and Clinical Characteristics

We assessed a positive correlation between AC and LFP measurements at baseline and over all the subsequent time-point evaluation follow-up (LFP baseline ρ 0.498, *p* < 0.001, LFP II ρ 0.460, *p* < 0.001, LFP III ρ 0.631, *p* < 0.001, LFP IV ρ 0.547, *p* = 0.006, LFP V ρ 0.767, *p* = 0.001) (Table 3 reports the evolution of LFP).

We observed significant correlations between the value of the LFP and the presence of complications at the different time points (LFP baseline ρ 0.360, *p =* 0.006, LFP II ρ 0.538, *p* < 0.001, LFP III ρ 0.410, *p* = 0.011), the presence of cataracts (LFP baseline ρ 0.542, *p* < 0.001, LFP II ρ 0.584, *p* < 0.001, LFP III ρ 0.523, *p* = 0.001, LFP IV ρ 0.452, *p =* 0.026), and the posterior synechiae (LFP II ρ 0.467, *p* = 0.001).

All the correlations performed are reported in Figure 4 and Appendix A.

Indeed, eyes which had complications showed higher LFP values compared to eyes which did not show complications at baseline (25.94 ± 33.31 vs. 10.01 ± 8.44, *p* < 0.001) (Figure 5), at the second measurement (22.044 ± 21.9 vs. 8.83 ± 4.62, *p* < 0.001), at the third measurement (22.077 ± 20.8 vs. 10.69 ± 5.56, *p* 0.003), at the fourth (21 ± 19.13 vs. 10.68 ± 6.19, *p* = 0.016), and at the last available follow-up too (30.33 ± 31.24 vs. 11.06 ± 8.07, *p* = 0.002).

## 4. Discussion

This prospective study is one of the few studies that assesses the role of LFP in routine clinical practice to evaluate ocular inflammation in children with uveitis in comparison with SLE. We identified a consistent positive correlation between the LFP values and SLE performed by an expert ophthalmologist. We demonstrated the applicability of this novel technique in a cohort of children, where it was able to distinguish children with uveitis from children without uveitis, and we used this technique to monitor ocular inflammation in an objective way. Additionally, we highlighted a positive relation between higher LFP values and the presence of structural complications.

Detection and monitoring of disease activity in uveitis is crucial for customizing the treatment of our children and preventing complications. The SUN grading system, which is the currently used referred system to evaluate ocular inflammation, shows several limitations. First of all, there is the subjective estimation of inflammation by an observer, even though there are clear rules to count the cells, and additionally the use of non-continuous variables does not allow estimation of minimal changes [5]. Novel technologies such as LFP and the anterior segment optical coherence tomography (AS-OCT) showed a good acceptability profile for children, but more importantly an objective ability to quantify ocular inflammation in the absence of an expert ophthalmologist [16,17,18,19]. As we evaluated in our study, the LFP values have a strong correlation with SLE performed by an expert ophthalmologist, and the technique is able to distinguish between patients with uveitis and without uveitis, and patients with active and inactive uveitis.

Indeed, uveitis leads to an alteration of the blood–aqueous barrier that changes the protein composition, and the LFP technique also offers the possibility to detect minimal changes in the composition of humor aqueous in chronic uveitis, even though there are not evident changes in the number of cells in the anterior chamber.

The first report of LFP use in this indication was published in 1972 by Herbort et al. [20], where they transformed flare from a qualitative to a quantitative measure of intraocular inflammation. Several clinical studies of uveitis patients have shown that flare measurements by LFP allow precise monitoring of uveitis, a response to therapy, and prediction of disease relapse or exacerbation. Correlations of LFP values with complications of uveitis and visual loss indicate that flare measurement by LFP is a valuable tool in a follow-up in patients with uveitis. However, there is limited evidence about the use of LFP, as far as we know, in childhood.

Thus, LFP offers an extremely objective instrument also able to detect minimal changes in the aqueous composition, with minimal collaboration from young patients compared to common SLE that, in in ambiguous situations, requires prolonged collaboration from young patients.

The use of this novel technique in routine clinical practice has already been reported in limited pediatric studies [8,10,13,15,21,22]. It was also recently used in association with SLE to monitor ocular inflammation in a randomized controlled trial conducted by Quartier et colleagues that contributed to the approval of adalimumab for the treatment of childhood chronic anterior uveitis [22]. Additionally, as recently reported by Yalcidang et al., it might be a useful tool for the monitoring of patients with idiopathic uveitis, as confirmed in our report where we assessed eleven patients with idiopathic uveitis that showed LFP parameters similar to JIA-associated uveitis and good concordance with AC grades over the time [13].

In accordance with recent studies, our study showed significantly increased LFP values in patients who showed structural complications such as cataracts and posterior synechiae, in a consistent way, regardless of inflammatory status [9,10,14,15,21,23,24]. In our cohort, we further showed a persistent correlation between LFP values and AC cells, even though there were complications. This seems in accordance with previous data reported not only in a large adult cohort such as Gonzales et al., but also in pediatric cohorts by Holland et al., Davis et al., and Yalcidang et al. [8,9,13,15].

However, our data seem to be in contrast with Tappeiner et al., who described an additional association between ocular hypertension and the epiretinal membrane, but a possible explanation might be that none of our patients showed these complications [14,20].

Additionally, we did not evaluate any correlation between the LFP values and visual acuity as evaluated by other authors such as Gonzales et al. [9].

However, caution should be posed when we compare our data with others, because we need to consider that our cohort of children showed a lower prevalence of complications and better visual outcomes compared to other cohorts [8,21]. Additionally, our study, in comparison to other cohorts, was not able to assess the predictive value of LFP in monitoring the course of uveitis and the occurrence of new complications, considering the different time points at which patients were included in this study [13].

Before drawing our conclusions, we need to point out several limitations of our study. First of all, we need to consider the small sample size of patients included, which did not allow us to draw firm conclusions, but we need to consider the rarity of the disease and the use of the novel technique. Additionally, the patients included in the study showed different anatomical subtypes. In particular, we included six patients with panuveitis at onset who, however, during follow-up mainly manifested an anterior involvement. Additionally, the aim of this study was to test the hypothesis that LFP is comparable to the standard SLE evaluation in a pediatric population. Thus, we were not able to consider a specific interval of evaluation. Most of the patients with uveitis underwent LFP during scheduled visits. Unfortunately, because of the exiguous number of patients included in this study and the different time points at which the patients were in their disease history, we were not able to determine a relationship between LFP and relapse and/or treatment.

Furthermore, we have variable follow-up for the different patients.

## 5. Conclusions

Laser flare photometry represents an optimal non-invasive, objective, and quantitative method to assess intraocular inflammation, with an acceptable profile for children. It showed a significant ability to distinguish patients with and without uveitis, with active and inactive uveitis, and with complications, with a significant correlation with SLE performed by an expert ophthalmologist. However, caution should be posed in the interpretation of these data, considering the sample size included in our study and short duration of follow-up. The increasing evidence about the use of this novel technique in childhood suggests enhancement of the use of LFP in the routine clinical ophthalmology visit, at least in tertiary centers, in combination with standard visits.

## Figures and Tables

**Figure 1 diagnostics-13-03179-f001:**
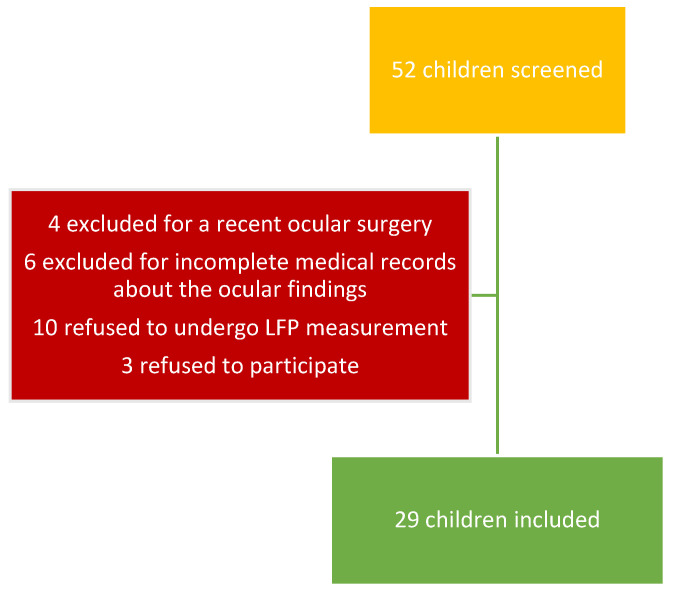
Selection process of the population included.

**Figure 2 diagnostics-13-03179-f002:**
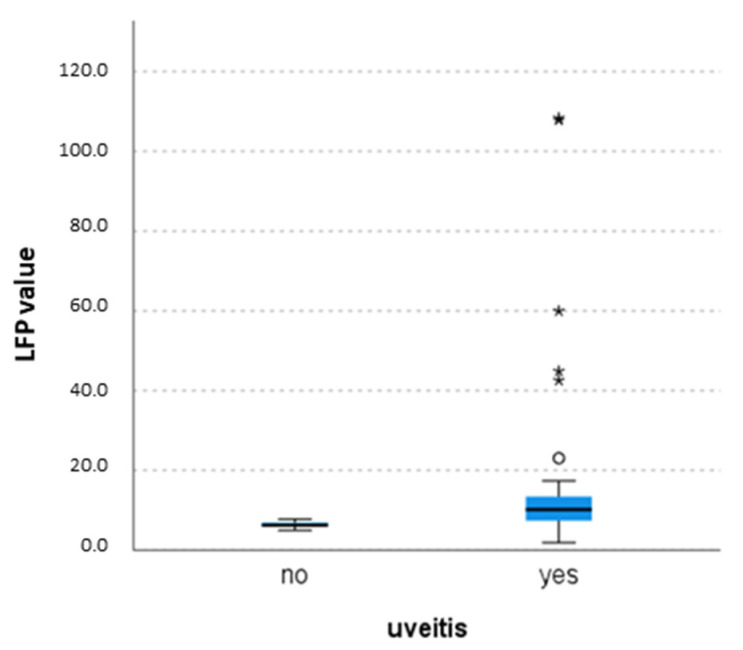
Comparison of laser flare photometry (LFP) values between patients with uveitis and without uveitis with the Kruskal–Wallis test (*p* = 0.007) that shows higher value of LFP values in patients with uveitis. The central line represents the distribution median, boxes span from the 25th to the 75th percentile, and error bars extend from the 10th to the 90th percentile. Asterixis are values higher than the 90th percentile.

**Figure 3 diagnostics-13-03179-f003:**
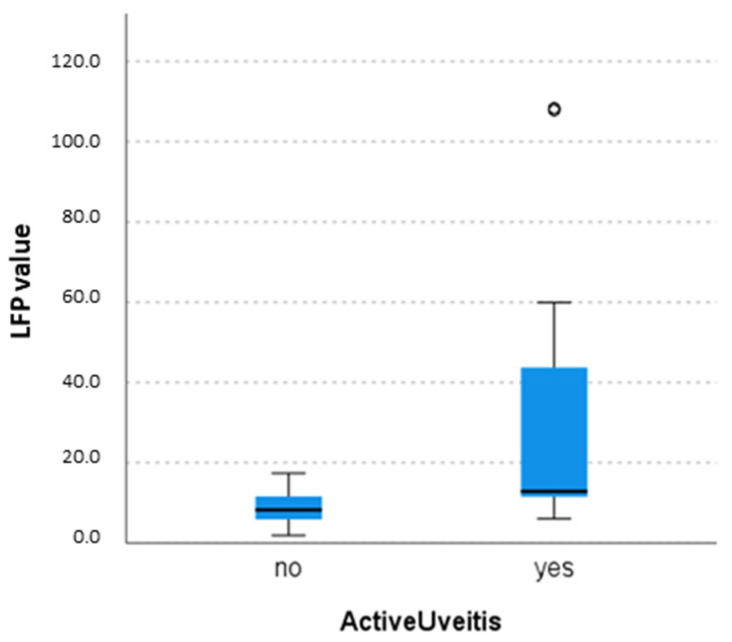
Comparison of laser flare photometry values between patients with active and inactive uveitis with the Kruskal–Wallis test (*p* = 0.002). The central line represents the distribution median, boxes span from the 25th to the 75th percentile, and error bars extend from the 10th to the 90th percentile. Dots are values higher than the 90th percentile.

**Figure 4 diagnostics-13-03179-f004:**
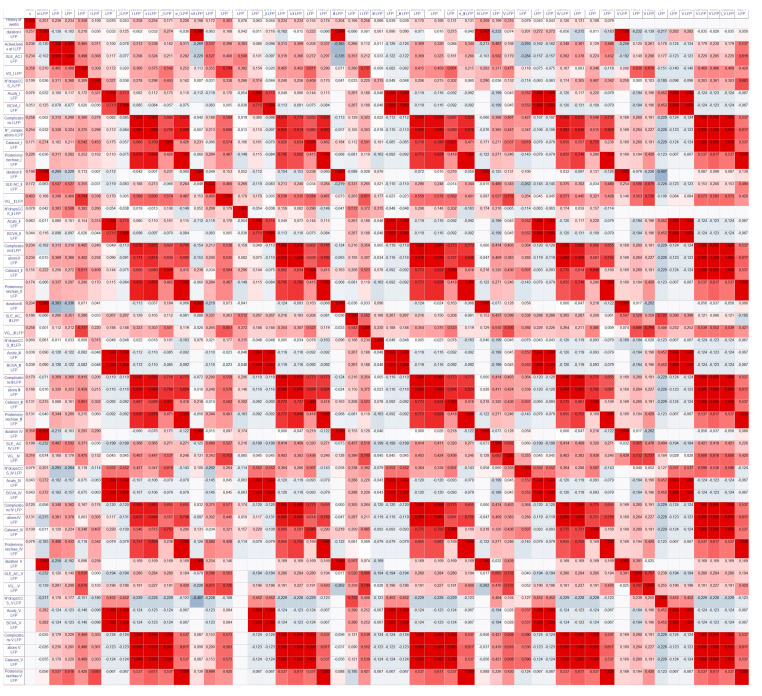
Chromatic covariation matrix of the correlations performed. The red color represents higher coefficient of correlation, while blue represents lower coefficient of correlation. List of abbreviations: SLE slit lamp evaluation, AC: anterior chamber cells, LFP laser flare photometry, N number, CCS Corticosteroid, VG LFP: value of LFP, Visual Acuity: visual acuity expressed in LogMar as a continuous variable, BCVA: best corrected visual acuity stratified as normal < 0.4, impaired 0.4–1, blindness > 1. The commas reported in the present figure should be considered as separator of decimal numbers.

**Figure 5 diagnostics-13-03179-f005:**
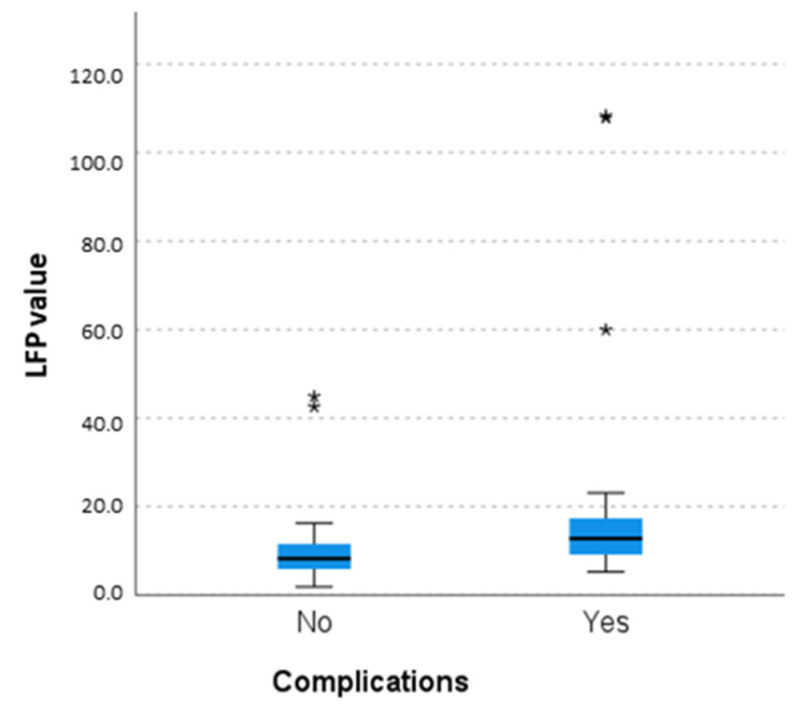
Comparison of laser flare photometry values in patients with uveitis with and without ocular complications (*p* < 0.001) that shows higher LFP values in patients with complicatios. The central line represents the distribution median, boxes span from the 25th to the 75th percentile, and error bars extend from the 10th to the 90th percentile. Asterixis are values higher than the 90th percentile.

**Table 1 diagnostics-13-03179-t001:** Clinical, laboratory, and ocular characteristics of the population at onset.

Variables	JIA-U (15 Patients)	JIA-No-U (3 Patients)	Idiopathic-U (11 Patients)	*p*-Value
**Sex, Female**	12 (80%)	2 (66.7%)	5 (45.5%)	Ns
**Age at onset, months, median (IQR)**	38 (24–77)	53 (32–53)	126 (62–144)	0.025 (Kruskal–Wallis test)
**Caucasian**	15 (100%)	3 (100%)	10 (90.9%)	Ns
**ANA (+)**	14 (93.3%)	3 (100%)	6 (54.5%)	χ^2^ 6.69, *p* = 0.035
**ANCA (+)**	3 (20%)	-	1 (9.1%)	Ns
**ESR onset, mm/h, median (IQR)**	31 (10.5–50.5)	11 (7–11)	15.5 (7.75–25.5)	Ns
**CRP onset, mg/dl, median (IQR)**	1.19 (0.29–2.47)	1.15 (0.6–1.15)	0.3 (0.06–0.55)	NS
**Bilateral uveitis**	14 (93.3%)	-	8 (72.7%)	Ns
**Anatomical Loc** **Anterior** **Intermediate** **Panuveitis**	15 (100%) - -	- - -	6 (54.5%) - 5 (45.5%)	Χ^2^ 8.44, *p =* 0.004
**Ocular Symptoms** **Pain** **Redness** **Photophobia** **Floaters/blurred vision**	2 (13.3%) 0 2 (13.3%) 00	-	9 (81.8%) 4 (36.4%) 9 (81.8%) 5 (45.5%) 2 (18.2%)	χ^2^ 13.87, *p* < 0.001 χ^2^ 7.2, *p* = 0.027 χ^2^ 13.12, *p* < 0.001 χ^2^ 6.25, *p* = 0.044 Ns
**VA in the worse eye, LogMar, mean (±SD)** **VA < 0.4** **VA 0.4–1**	0.07 (0.27) 12 (80%) 1 (6.7%)	0 (0) 3 (100%) -	0.05 (0.13) 8 (72.7%) 1 (9.1%)	Ns
**VA R LogMar, mean (±SD)** **VA < 0.4** **VA 0.4–1**	0 (0) 13 (86.7%) -	0 (0) 3 (100%) -	0.04 (0.11) 8 (72.7%) 1 (9.1%)	Ns
**VA L LogMar, mean (±SD)** **<0.4** **0.4–1**	0.07 (0.27) 12 (80%) 1 (6.7%)	0 (0) 3 (100%) -	0.02 (0.07) 9 (81.8%) -	NS
**Presence of complications onset, n (%)**	2 (13.3%)	-	8 (72.7%)	χ^2^ 13.95, *p* < 0.001
**N of complications onset, median (IQR)**	0 (0–0)	0 (0–0)	2 (1–4)	<0.001 (Kruskal–Wallis test)

List of abbreviations: IQR interquartile range, CRP C-reactive protein, ESR Erythro-sedimentation rate, loc location, R right, L left, LFP laser flare photometry, m months, VA visual acuity, n number, ANA antinuclear antibody, ANCA Anti-neutrophil cytoplasmic antibody.

**Table 2 diagnostics-13-03179-t002:** Comparison of clinical and ocular characteristics in patients with uveitis and without uveitis at the time of the first laser flare photometry measurement.

Variables in Study	JIA-U (15 pts/30 E)	Idiopathic-U (11 pts/22 E)	JIA-No-U (3 pts/6 E)	*p*-Value U vs. No-U
**Age at baseline LFP median (IQR), months**	140 (105–167)	163 (143–188)	126.5 (94–126)	NS
**Duration of disease at baseline LFP, median (IQR)**	66 (34–121)	18 (14–60)	27 (13–41)	0.05
**Baseline LFP value for eye, mean (SD), ph/ms**	19.2 (±26.03)	10.95 (±11.19)	6.2 (±1.01)	0.007
**Baseline AC cell grade, mean (SD)**	0.417 (±0.813)	0.18 (±0.45)	0 (±0)	Ns
**Baseline Visual acuity, mean (SD) LogMar**	0.001 (±0.007)	0..0318 (0.14)	0 (±0)	ns
**Presence of complications, n (%)**	8 E (26.7%)	9 E (40.9%)	0	χ^2^ 3.84, *p* 0.05
**Cataracts, n (%)**	8 (26.7%)	1 (4.5%)	0	Ns
**Posterior synechiae, n (%)**	6 (20%)	8 (36.4)	0	Ns

List of abbreviations: U uveitis, JIA juvenile idiopathic arthritis, IQR interquartile range, LFP laser flare photometry, SD standard deviation, AC anterior chamber, n number, pts patients, E eyes, vs. versus.

**Table 3 diagnostics-13-03179-t003:** This table reports the evolution of the mean of LFP values and mean of slit lamp at the different time points considered, with the different complications evaluated.

Variables in Study	I LFP Measure (52 Eyes)	II LFP Measure (48 Eyes)	III LFP Measure (38 Eyes)	IV LFP Measure (24 Eyes)	V LFP Measure (16 Eyes)
**Duration of disease, months, median (IQR)**	51 (16.5–101.2)	61 (29–123)	72 (35–132.5)	62.5 (32–77)	71.5 (33.5–99.75)
**LFP value, ph/ms, mean (SD)**	14.73 (±20.3)	13.79 (±15.12)	14.55 (±13.8)	13.26 (±11.34)	14.6 (±15.5)
**SLE-AC cell grade, mean (SD)**	0.29 (±0.66)	0.219 (±0.43)	0.705 (±2.55)	0.33 (±0.48)	0.43 (±0.17)
**Visual acuity LogMar, mean (SD)**	0.01 (±0.09)	0.015 (±0.1)	0.018 (±0.11)	0.029 (±0.14)	0.04 (±0.17)
**Presence of complications, n eyes (%)**	17 (29.3%)	18 (37.5%)	13 (34.2%)	6 (25%)	3 (18.8%)
**Cataracts,** **n eyes (%)**	9 (15.5%)	10 (20.8%)	9 (23.7%)	6 (25%)	3 (18.8%)
**Posterior synechiae,** **n eyes (%)**	14 (24.1%)	12 (25%)	9 (23.7%)	3 (12.5%)	1 (6.3%)

List of abbreviations: SD standard deviations, pts patients, E eye, IQR interquartile range, LFP laser flare photometry, n number.

## Data Availability

Data will be available upon reasonable request to the corresponding author.

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
