# Peer review of "Laser Flare Photometry to Monitor Childhood Chronic Uveitis: A Preliminary Report of a Monocentric Italian Experience"

_diagnostics, 2023, doi:10.3390/diagnostics13203179_

Round 1
Reviewer 1 Report
I would like to congratulate the authors on the interesting topic and study. I would like them to clarify the following topics.
1. Where there any additional criteria used in the exclusion criteria mentioned in the methods section? For example, was the use of steroid eye drops, or cyclopentolate drops permitted prior to examination? Was recent (e.g. less than 3 months) surgery permitted?
2. In the statistical analysis section you correctly state that you used parametric or non-parametric tests as deemed appropriate. Please add the method of the evaluation of normality of distribution of the variables analyzed.
3. I would suggest to add a figure to explain how did you arrive at the study population of 29 patients included in the study and add how many patients declined to participate, or were deemed ineligible to participate due to exclusion criteria.
1.Please delete final s in "the cornerstones" in abstract.
2. Please rephrase ", since, as not timely and properly recognized and treated, it can lead to..." in backgroud.
3. Please add thee word "it" before is crucial in: " In order to prevent these complications and therefore blindness is crucial to start" to change to "In order to prevent these complications and therefore blindness,it is crucial to start...".
4. Please use right form in "the number of cells presents in the anterior chamber" and change it to "the number of cells present in the anterior chamber".
5.Please rephrase this "to have less than 16 years old at the moment of the LFP assessment" in methods section.
6. Please change "characteristic" to characteristics in the study procedures section.
7. Please change fluorangiography to fluorescein angiography in the study procedures section.
8. Please change "We observed a significant correlations" to significant correlation.
Author Response
I would like to congratulate the authors on the interesting topic and study. I would like them to clarify the following topics.
We thank the reviewer 1 for the positive comments about our study. Here a response point by point about his/her comments:
1. Where there any additional criteria used in the exclusion criteria mentioned in the methods section? For example, was the use of steroid eye drops, or cyclopentolate drops permitted prior to examination? Was recent (e.g. less than 3 months) surgery permitted?
We thank the reviewer about this comment because allow us to clarify that the measurements of patients who underwent to surgery were not included if the surgery was performed within 3 months from surgery. Topical corticosteroid and cyclopentolate were permitted.
2. In the statistical analysis section you correctly state that you used parametric or non-parametric tests as deemed appropriate. Please add the method of the evaluation of normality of distribution of the variables analyzed.
We thank the reviewer for this comment, that allow us to specify how we evaluated the distribution of the analysed variables. We added this sentence in the statistical section “Q-Q plots were used to test the assumption of normality.”
3. I would suggest to add a figure to explain how did you arrive at the study population of 29 patients included in the study and add how many patients declined to participate, or were deemed ineligible to participate due to exclusion criteria.
We thank the reviewer for this comment, as you suggested we added a figure in order to explain the final samples size of our study.
Comments on the Quality of English Language
1. Please delete final s in "the cornerstones" in abstract.
Thanks for your comment, we modified accordingly.
2. Please rephrase ", since, as not timely and properly recognized and treated, it can lead to..." in backgroud.
Thanks for your suggestion, we modified the sentence as follow “Chronic non-infectious uveitis is a challenging disease from a diagnostic and treatment perspective, due to the risk of severe complications, including blindness, if it is not promptly and correctly identified and managed”
3. Please add thee word "it" before is crucial in: " In order to prevent these complications and therefore blindness is crucial to start" to change to "In order to prevent these complications and therefore blindness, it is crucial to start...".
Thanks for your comment, we added it as you suggested.
4. Please use right form in "the number of cells presents in the anterior chamber" and change it to "the number of cells present in the anterior chamber".
Thanks for your comment, we modified it as you suggested.
5. Please rephrase this "to have less than 16 years old at the moment of the LFP assessment" in methods section.
Thanks for your comment, we modified the sentence as follows “to be ≤ 16 years old when LFP was performed”.
6. Please change "characteristic" to characteristics in the study procedures section.
Thanks for your comment, we modified the text accordingly to your suggestion.
7. Please change fluorangiography to fluorescein angiography in the study procedures section.
Thanks for your comment, we modified the text accordingly to your suggestion.
8. Please change "We observed a significant correlations" to significant correlation.
Thanks for your comment, we modified the text accordingly to your suggestion.
Reviewer 2 Report
I have read the paper by dr Maccora et al. in which the authors aimed to compare the slit-lamp examination (SLE) and laser flow photometry (LFP) techniques in the assessment of anterior chamber inflammation in pediatric uveitis.
The first report of LFP use in this indication was published in 1972 by Herbort et al. (Herbort CP, Guex-Crosier Y, de Ancos E, et al. Use of laser flare photometry to assess and monitor inflammation in uveitis. Ophthalmology. 1997;104:64–71). LFP allows to assess the amount and size of proteins in the aqueous humour and represents diagnostic progress, transforming flare from a qualitative to a quantitative measure of intraocular inflammation. LFP allows detection of subclinical alterations in the blood-ocular barriers, identifying subtle changes that could not have been recorded otherwise. Many clinical studies of uveitis patients have shown that flare measurements by LFP allow precise monitoring of uveitis; monitoring the response to therapy and predict the disease relapse or exacerbation. Correlations of LFP values with complications of uveitis and visual loss indicate that flare measurement by LFP is a valuable tool in a follow-up in patients with uveitis.
Methodology: There are not strictly planned intervals between follow-ups thus it is imposiible to conduct precise observation and changes in LFP velues in analysed group of patients. Did the authors were able to predict the relapse or exacerbation of intraocular inflammation beased on LFP measurements ? Did the authors observed a relationship between reduction of steroids dosing / immunosuppressive therapy and LFP measurement?
Results: The paper include 3 tables which are not prepared thoroughly and which make the reader confused.
Figure 1 is not necessary as the same data are included in table 1.
English needs to be improved.
In the references list the authors do not use the standard abbrevations of cited journals.
All my comments have been entered in the attached PDF file at the margins as comments „bubbles”.
In my opinion the paper is not innavative and it does not bring any new observations on the LFP application in ophthalmology.

English needs to be improved.
e.g. lose of acuity; should be: a visual loss
"Before drawing our conclusions, we need several limitations of our study" - I think that the authors aimed to point out the limitations of the study.
Author Response
REVIEWER 2 Comments and Suggestions for Authors
I have read the paper by dr Maccora et al. in which the authors aimed to compare the slit-lamp examination (SLE) and laser flow photometry (LFP) techniques in the assessment of anterior chamber inflammation in pediatric uveitis.
The first report of LFP use in this indication was published in 1972 by Herbort et al. (Herbort CP, Guex-Crosier Y, de Ancos E, et al. Use of laser flare photometry to assess and monitor inflammation in uveitis. Ophthalmology. 1997;104:64–71). LFP allows to assess the amount and size of proteins in the aqueous humour and represents diagnostic progress, transforming flare from a qualitative to a quantitative measure of intraocular inflammation. LFP allows detection of subclinical alterations in the blood-ocular barriers, identifying subtle changes that could not have been recorded otherwise. Many clinical studies of uveitis patients have shown that flare measurements by LFP allow precise monitoring of uveitis; monitoring the response to therapy and predict the disease relapse or exacerbation. Correlations of LFP values with complications of uveitis and visual loss indicate that flare measurement by LFP is a valuable tool in a follow-up in patients with uveitis.
We thank the reviewer to point out the importance of the use of LFP for the assessment of uveitis. Thus we added this valuable comment into the Discussion of the manuscript.
Methodology: There are not strictly planned intervals between follow-ups thus it is impossible to conduct precise observation and changes in LFP values in analysed group of patients. Did the authors were able to predict the relapse or exacerbation of intraocular inflammation based on LFP measurements ? Did the authors observed a relationship between reduction of steroids dosing / immunosuppressive therapy and LFP measurement?
We thank the reviewer for this valuable comments. Aim of this study was to test the hypothesis that LFP is comparable to the standard SLE evaluation in a pediatric population. Thus we were not able to consider a specific interval of evaluation. Most of the patients with uveitis have undergone to LFP during scheduled visits. Unfortunately, because of the exiguous number of patients included in this study and the different time points in whom the patients were in their disease history, we were not able to determine a relationship between LFP and relapse and/or treatment. This will be the aim of the next study that will include a bigger sample and a longer follow-up. We added this point as limitation of the study.
Results: The paper include 3 tables which are not prepared thoroughly and which make the reader confused.
We thank the reviewer for this comment, we reviewed the tables as you suggested in order to improve the comprehension of them.
Figure 1 is not necessary as the same data are included in table 1.
We thank the reviewer for the comment, we removed the figure 1 as requested.
English needs to be improved.
We thank the reviewer for the comment, our manuscript was now reviewed by a native English speaker.
In the references list the authors do not use the standard abbrevations of cited journals.
We thank the reviewer for the suggestion, we reviewed all the references accordingly to your and Editor’s comments.
All my comments have been entered in the attached PDF file at the margins as comments „bubbles”.
We thank the reviewer for all the comments across the PDF, we addressed all the comments. Below are reported additional replies to his/her comments in the following response.
In my opinion the paper is not innavative and it does not bring any new observations on the LFP application in ophthalmology.
As the reviewer pointed out this paper did not report new applications of LFP in ophthalmology and we discuss this point into the discussion. However, this manuscript might add evidence about the use of this technique into the childhood non-infective chronic uveitis. As the reviewer perfectly knows, there are no specific data regarding the use of LFP into the paediatric age. There are just very few papers reporting the use of LFP in childhood, thus we think that our manuscript, although the caveats that the reviewer properly pointed out, might add evidence regarding the need of use this technique even into the pediatric setting. Complications in table 2 are not the same reported in table 1.
We thank the reviewer for this comment that allow us to clarify this difference in the number and type of complications. As you may understand in table 1 we reported the complications at the onset of uveitis, while in table 2 we reported the complications at the moment of the first evaluation with LFP. Unfortunately, all the measurement with LFP were performed not at the time of uveitis diagnosis, but during the disease course, so several complications were already resolved.
Age at baseline LFP measurement: please delete
We thank the reviewer for this comment. This datum does not represent the age at the disease onset, rather the age of the patients at the time of the first LFP. This information might result helpful to let the readers know the younger child performed LFP.
English needs to be improved. e.g. lose of acuity; should be: a visual loss
We thank the reviewer for the comment we modified the text accordingly to the suggestion.
"Before drawing our conclusions, we need several limitations of our study" - I think that the authors aimed to point out the limitations of the study.
Thanks for your comment, as you understood we would like to point out the limitations of the study, unfortunately there was a typo that now we addressed.
Reviewer 3 Report
The paper is interesting. However, some points need to be discussed in revision.
1 Previous articles have been published on the use of Laser Flare Photometry in chronic non-infectious uveitis. (YalçındaÄŸ FN, Köse HC, Temel E. Comparative study of laser flare photometry versus slit-lamp cell measurement in pediatric chronic non-infectious anterior uveitis. Eur J Ophthalmol. 2023 Jan;33(1):382-390. )Please compare the differences or innovations in this article.
2 Figure 1: “3 with JIA without uveitis (10.3%), 15 with JIA-uveitis (51.7%), and 11 with idiopathic chronic uveitis (37.9%) (Figure 1 )”(last paragraph of Page 3), However, we only can find 10( 4 anterior uveitis + 6 panuveitis) idiopathic chronic uveitis in the figure 1(panuveitis)?
3 The small sample size of patients (15 with JIA-uveitis ) limits the study, making statistical conclusions prone to bias and decreased feasibility.
Author Response
REVIEWER 3
The paper is interesting. However, some points need to be discussed in revision.
We thank the reviewer 3 for his/her positive comment, herein a response point by point to your comments.
1 Previous articles have been published on the use of Laser Flare Photometry in chronic non-infectious uveitis. (YalçındaÄŸ FN, Köse HC, Temel E. Comparative study of laser flare photometry versus slit-lamp cell measurement in pediatric chronic non-infectious anterior uveitis. Eur J Ophthalmol. 2023 Jan;33(1):382-390. )Please compare the differences or innovations in this article.
We thank the reviewer for this comment, we address this specific point in discussion adding a paragraph. “Additionally, as recently reported by Yalcidang et al it might be a useful tool for the monitoring of patients with idiopathic uveitis as confirmed in our report where we assessed eleven patients with idiopathic uveitis that showed LFP parameters similar to JIA associated uveitis and good concordance with AC grades over the time”, “As previously mentioned the value of the LFP is the result of the protein concentration of the humor aqueous, nevertheless, when specific complications are present in the eyes of the patients this value dramatically increases as demonstrated in several study 9,10,14,15,20,22,23.
In accordance with these findings we evaluated significant increased LFP values in patients who showed structural complications as cataract and posterior synechiae, in a consistent way, regardless of inflammatory status.”. Additionally, our study in comparison to other cohorts was not able to assess predictive value of LFP in monitoring the course of uveitis and the occurrence of new complications considering the different time point in which the patients were included in this study13.
2 Figure 1: “3 with JIA without uveitis (10.3%), 15 with JIA-uveitis (51.7%), and 11 with idiopathic chronic uveitis (37.9%) (Figure 1 )”(last paragraph of Page 3), However, we only can find 10( 4 anterior uveitis + 6 panuveitis) idiopathic chronic uveitis in the figure 1(panuveitis)?
We thank the reviewer for this comment, that allow us to correct this type, as you understood the patients with idiopathic uveitis that showed a panuveitis are 5 rather 4. We corrected the figure and table.
3 The small sample size of patients (15 with JIA-uveitis ) limits the study, making statistical conclusions prone to bias and decreased feasibility.
We thank the reviewer for this comment as you highlighted this is one of the limitations of our study. We added a specific comment about this in the limitations of the study section.
After this pilot study we aim to prospectively increase the sample size of our cohort.
However, at the present, our manuscript reporting the use of LFP in childhood represents one of the first attempts to routinary use this technique in a proper evaluation of a child suffering from chronic non infective uveitis.
Reviewer 4 Report
Thank you for your trust and entrusting me with the role of reviewer. Diagnosis and treatment of uveitis in children is a big problem. The paper describes an innovative diagnostic method, laser flare photometry, which is helpful in assessing inflammation in the anterior segment of the eye and possible complications of the disease. The research is well documented, but as the authors themselves emphasize, the size of the groups is small. This would indicate the need for continued research. Unfortunately, not all centers have such possibilities. The statistical analysis used raises no doubts. References cited correctly.
Author Response
REVIEWER 4
Thank you for your trust and entrusting me with the role of reviewer. Diagnosis and treatment of uveitis in children is a big problem. The paper describes an innovative diagnostic method, laser flare photometry, which is helpful in assessing inflammation in the anterior segment of the eye and possible complications of the disease. The research is well documented, but as the authors themselves emphasize, the size of the groups is small. This would indicate the need for continued research. Unfortunately, not all centers have such possibilities. The statistical analysis used raises no doubts. References cited correctly.
We are extremely grateful to the reviewer 4 for these positive comments, and as she/he emphasized that research in this field is needed to enable widespread use of this new technique.
Round 2
Reviewer 2 Report
Thanks to the authors for responding to all my remarks and comments. All answers and explanations are comprehensive. Tables were revised and now it’s more easy to understand them.
A negative feature of this work is its methodology – as authors underlined - they
showed a variable follow-up for the different patients and different intervals between
follow-ups. I believe that authors will continue their study including a greater number
of patients, improving the methodology and thus they will shed a new perspective
on the application of laser flare photometry in the diagnosing and monitoring
the paediatric uveitis.
Maybe its worth to consider some minor changes in a title of a manuscript: Laser Flare
Photometry to Monitor Childhood Chronic Uveitis: A Preliminary Report
of a Monocentric Italian Experience.
English still needs some minor improvements.
English still needs some minor improvements.
Author Response
Reviewer 2:
- Thanks to the authors for responding to all my remarks and comments. All answers and explanations are comprehensive. Tables were revised and now it’s more easy to understand them. A negative feature of this work is its methodology – as authors underlined – they showed a variable follow-up for the different patients and different intervals between follow-ups. I believe that authors will continue their study including a greater number of patients, improving the methodology and thus they will shed a new perspective on the application of laser flare photometry in the diagnosing and monitoring the paediatric uveitis. Maybe its worth to consider some minor changes in a title of a manuscript: Laser Flare Photometry to Monitor Childhood Chronic Uveitis: A PreliminaryReport of a Monocentric Italian Experience.
We thank the reviewer 2 for the comments, as you suggested we changed the title of our manuscritp.
- English still needs some minor improvements.
We thank the reviewer 2 for this comment, we reviewed again our manuscript for English language.
Reviewer 3 Report
The paper is interesting. I suggest it would be published in the journal.
Author Response
Reviewer 3:
The paper is interesting. I suggest it would be published in the journal.
We thank reviewer 3 to endorse our paper.